# Is Post-Transplant Cyclophosphamide the New Methotrexate?

**DOI:** 10.3390/jcm10163548

**Published:** 2021-08-12

**Authors:** Alberto Mussetti, Annalisa Paviglianiti, Rocio Parody, Anna Sureda

**Affiliations:** 1Clinical Hematology Department, Institut Català d’Oncologia-Hospitalet, 08908 Barcelona, Spain; annalisapaviglianiti@iconcologia.net (A.P.); rparody@iconcologia.net (R.P.); asureda@iconcologia.net (A.S.); 2Institut d’Investigació Biomèdica de Bellvitge (IDIBELL), 08908 Barcelona, Spain; 3Medicine Department, Universitat de Barcelona (UB), 08007 Barcelona, Spain

**Keywords:** allogeneic transplantation, GVHD, post-transplant cyclophosphamide, HLA-identical, haploidentical

## Abstract

Introducing post-transplant, cyclophosphamide (PT-Cy) graft-versus-host disease (GVHD) prophylaxis in the setting of haploidentical donor transplantation has marked the most important advance in allogeneic hematopoietic cell transplantation (alloHCT) within the past 15 years. The efficacy of this procedure and its simple features have allowed for the significantly widespread application of alloHCT worldwide. Indeed, the procedure’s effectiveness in reducing immunological complications in the haploidentical setting has even challenged the status quo use of calcineurin-inhibitor, methotrexate-based GVHD prophylaxis in the setting of HLA-identical donors. Currently, however, prospective clinical trials in support of PT-Cy-based GVHD prophylaxis in the HLA-matched setting are striving to resolve the matter of its potential role. This review will briefly report the overall outcomes of PT-Cy-based GVHD prophylaxis in the haploidentical setting and summarize results obtained in the HLA-identical field. We will present future perspectives at the end of the manuscript.

## 1. Introduction

HLA mismatches have always represented a major barrier to successful alloHCT. However, toxicity-related mortality (TRM) of alloHCT has decreased in the last 30 years. We can attribute the decline to the discovery of the HLA complex; the creation of international donor registries to identify HLA-matched donors; and improvements in GVHD prophylactic strategies. Yet, when patients lack an HLA-identical donor—related or unrelated—the use of alternative donors such as haploidentical ones or cord blood units becomes the only strategy available [1].

When clinicians used a haploidentical donor, different immunosuppressive strategies such as T cell depletion underwent testing [2]. However, the risks related to immunological complications or infection-related toxicity often exceeded the benefits of an alloHCT. Post-transplant cyclophosphamide was introduced into clinical practice at the beginning of the 21st century as a simpler and effective strategy for haploidentical alloHCT [3]. From a biological standpoint, its efficacy is related to a strong induction of immune tolerance via peripheral and central mechanisms [4,5].

Following initial proof-of-concept studies that demonstrated the feasibility and safety of this strategy, several retrospective analyses have revealed how results obtained from haploHCT by PT-Cy could compete with standard calcineurin-inhibitor, methotrexate-based GVHD strategies for matched-unrelated donor (MUD) alloHCT [4,5]. Historically, methotrexate has been considered the most optimal companion to calcineurin inhibitors in the setting of HLA-identical donors. In a randomized trial performed in the 1980s, Storb et al. showed that cyclosporine plus methotrexate was superior to cyclosporine alone [6]. In a later setting of MUD donors, the same was shown for tacrolimus and methotrexate, with similar results between the two calcineurin inhibitors [7,8]. The past 20 years have also borne witness to the emergence of in vivo T cell depletion by means of polyclonal anti-thymocyte globulin (ATG) serum as an effective strategy in lowering both acute and chronic GVHD incidence, especially in the MUD setting. At least four randomized clinical trials—of which one used rabbit ATG and the other three used anti-T-lymphocyte globulin—were successful in showing the superiority of T cell depletion to calcineurin inhibitors and methotrexate alone [9,10,11,12]. Currently, PT-Cy competes against methotrexate with or without ATG with respect to becoming the ideal complement to calcineurin inhibitors in the setting of both related and unrelated donors. Strong evidence in support of PT-Cy is, however, lacking. Most data come from either small-sized or retrospective studies. However, prospective trials are ongoing, and several centers have already applied this strategy in the HLA-matched setting as well. Thus, a critical reappraisal of the literature is necessary at the moment, as clinicians await evidence-based answers to the matter-at-hand.

## 2. Methods

We reviewed a PubMed and Cochrane database search using the terms “post-transplant cyclophosphamide”, “allogeneic hematopoietic cell transplant”, “haploidentical donor”, and “matched donor”. We applied peer-reviewed journals, human studies, and the English language to the filter search. We performed additional searches to identify articles related to topics pertaining to each discussion section. We read all identified articles completely, extracting and summarizing relevant information. We made the study selection based on prioritization of the type of study (prospective over retrospective), population size, and date of publication.

## 3. Haploidentical Setting

The first clinical study of PT-Cy in haploHCT was conducted in the setting of non-myeloablative conditioning (Hopkins’ protocol), with subsequent clinical trials concluding PT-Cy be at a dose of 50 mg/kg on days +3 and +4, and a combination of mycophenolate mofetil (MMF) and calcineurin inhibitor (CNI) on day +5 [3]. Since then, increasing worldwide experiences have been reported in retrospective studies and across different hematologic diseases, with overall similar outcomes when compared with those of standard donors (Table 1). Surprisingly, with respect to the classical CNI-based prophylaxis used with matched-related donors (MRD) and MUD, PT-Cy appeared to be associated with a significantly lower rate of chronic GVHD. This effect could, however, be attributed to the more frequent use of bone marrow for haploidentical HCT in the beginning. For the first-time ever, Ciurea et al. compared clinical outcomes of haploidentical HCT with PT-Cy versus MUD to standard CNI-based GVHD prophylaxis [5]. Despite no differences in overall survival, there was less incidence of chronic GVHD in favor of haploidentical HCT. Kanate et al. reported similar results in the lymphoma setting and reduced-intensity conditioning regimen [4]. With the use of registry data, haploidentical HCT was shown to have comparable survival rates to those of MUD. The incidence of chronic GVHD was lower for the haploidentical HCT group as well.

Additionally, conditioning intensity has evolved from non-myeloablative to myeloablative regimens over the years. The classical Baltimore protocol, which finds it basis in the combination of fludarabine-cyclophosphamide and total body irradiation (200cGy), was associated with a low TRM yet high relapse rates. The pursuit for more intensive conditioning schemes has, therefore, gained traction. In this setting, increasing experience in the past decade has grown support for the use of myeloablative regimens with tolerable, non-relapse mortality for younger patients and in high-risk diseases. The most frequently used regimens at the moment are based on alkylating drugs such as thiotepa with either busulfan (a total dose of 6.4–9.6 mg/kg intravenously), melphalan (140 mg/m^2^), or fludarabine (150 mg/m^2^). There is a need for individualizing the conditioning regimen in accordance with basal disease risk to counterbalance non-relapse mortality and relapse risk.

## 4. Marrow versus Peripheral Blood Grafts within the PT-Cy Platform

The first clinical trials in the setting of T-cell-replete haploidentical HCT with PT-Cy systematically used bone marrow as a source, although peripheral blood stem cells have become the predominant source for alloHCT in adults worldwide. Clinical advantages of peripheral blood stem cells, when compared to bone marrow, include ease in collection, faster hematologic recovery, and a lower risk of graft failure and relapse [21,22]. A main concern about peripheral blood stem cells is the high number of donor T cells present in the peripheral blood and the potentially increased risk of GVHD. As a result, this issue resulted in establishing bone marrow as the first choice for T-cell-replete haploidentical HCT.

However, consecutive studies have shown that peripheral blood stem cells can be safely used in this setting, with acceptable acute and chronic GVHD incidences similar to those observed after MRD and MUD with peripheral blood stem cells [23,24,25]. Controversy, nonetheless, remains and has been reported in the most relevant retrospective studies across international registries.

A retrospective study with 681 patients compared transplant outcomes after non-myeloablative, T-cell-replete haploidentical HCT using peripheral blood stem cells versus bone marrow. The findings revealed similar hematopoietic recovery, non-relapse mortality, and 2-year overall survival in both groups. Conversely, investigators observed a higher incidence of acute and chronic GVHD, as well as a lower risk of relapse after haploidentical HCT with peripheral blood stem cells [26]. Remarkably, the difference reported in relapse rates applied to acute myeloid leukemia but not lymphoma. A retrospective study focusing on Hodgkin lymphoma showed better GVHD-free, relapse-free survival, overall survival, and progression-free survival (PFS) for peripheral blood stem cells [27]. Similarly, a retrospective and multicenter study from the European Society for Blood and Marrow Transplantation compared bone marrow and peripheral blood stem cells in 451 patients with either acute myeloid leukemia or acute lymphoblastic leukemia who underwent haploidentical HCT [28]. No differences in chronic GVHD, relapse rate, non-relapse mortality, overall survival, and leukemia-free survival were found between both groups. Bone marrow recipients had a lower engraftment rate (92% vs. 95%, *p* < 0.001) and decreased incidence of grade 2–4 acute GVHD (21% vs. 38%, *p* ≤ 0.01; respectively). In the multivariate analysis, peripheral blood stem cells were associated with an increased risk of acute GVHD. More recent data from the Center for International Blood and Marrow Transplant Research analyzed overall outcomes of PT-Cy haploidentical HCT in adult patients with acute myeloid leukemia, acute lymphoblastic leukemia, myelodysplasia, and chronic myeloid leukemia. Patients were categorized into four groups based on conditioning intensity and stem cell source: MAC-bone marrow (#79); MAC-peripheral blood stem cells (#183); reduced-intensity (RIC)-bone marrow (#192); and RIC-peripheral blood stem cells (#192) [29]. Significant differences were found in grade 2–4 acute and 1-year chronic GVHD in the univariate analysis, with MAC-peripheral blood stem cells being the highest (44% and 40% respectively), followed by RIC-peripheral blood stem cells (36% and 34%), MAC-bone marrow (36% and 24%), and RIC-bone marrow (30% and 20%). In the multivariable analysis, there was no impact of stem cell source or conditioning regimen on grade 2–4 acute GVHD. In contrast, peripheral blood stem cells were a significant risk factor for the development of chronic GVHD in the RIC setting. There were no differences in relapse rate or survival between groups.

In conclusion, considering the biases of retrospective studies, both bone marrow and peripheral blood stem cells can be used safely in the setting of PT-Cy haploidentical HCT. However, peripheral blood stem cells can be associated with a higher risk of acute and chronic GVHD in certain patient groups. Further investigation is needed to identify the best strategy in accordance with basal disease and conditioning. 

## 5. HLA-Identical Setting (MUD, MRD, Mismatched-UD)

A list of relevant studies in the non-haploidentical setting is reported in Table 2.

In the MAC-bone marrow setting, Luznik et al. were the first to use only PT-Cy for high-risk hematologic patients receiving a bone marrow-derived graft [30]. Both MRD and MUD were used in this setting. Grade 2–4 and 3–4 acute GVHD were 43% and 10%, respectively, at day +100. Two-year all-grade chronic GVHD was approximately 10%. Acceptable 2-year non-relapse mortality, event-free survival, and overall survival were reported at 17%, 39%, and 59%, respectively. A similar study from Kanakry et al. using busulfan/fludarabine MAC conditioning instead of busulfan/cyclophosphamide reported almost similar results of grade 2–4 and grade 3–4 acute GVHD of 51% and 15%, respectively, at day +100 [31]. One-year non-relapse mortality, 2-year disease-free survival, and overall survival were 16%, 62%, and 67%, respectively. The same author showed in a larger cohort of 209 adult patients with heterogeneous diseases who underwent MAC using bone marrow graft that grade 2–4 acute GVHD was 45% at day +100. Three-year non-relapse mortality, disease-free survival, and overall survival were 17%, 46%, and 58%, respectively [32].

**Table 2 jcm-10-03548-t002:** Selection of prospective studies using PT-Cy in the setting of HLA-matched donor alloHCT. GVHD = graft versus-host disease; MAC = myeloablative conditioning; PT-Cy = post-transplant cyclophosphamide; MRD = matched related donor; MUD = matched unrelated donor; BM = bone marrow; Bu = busulfan; Flu = fludarabine; TBI = total body irradiation; CSA = cyclosporine; PBSC = peripheral blood stem cell; MMF = mycophenolate mofetil; Treo = treosulfan; MMUD = mismatched unrelated donor; CNI = calcineurin inhibitors; ATG = anti-thymocyte globulin.

Study	Type of Conditioning	GVHD Prophylaxis	Donor	Graft	Acute GVHD	Chronic GVHD	Overall Survival	Commentary
**Luznik** [30]	MAC (Bu/Cy)	PT-Cy day +3, +4	MRD (#78) MUD (#39)	BM	Grade 2–4 43%	9% and 11% for MRD and MUD, respectively	55% at 2 years	First study to prove feasibility of PT-Cy in the HLA-matched setting
**Mielcarek** [33]	MAC(Bu/Flu, #25; TBI#18)	PT-Cy day +3, +4 and CSA	MRD (#12) MUD (#31)	PBSC	Grade 2–4 77%	16%	70% at 2 years	This study showed that when using MAC and PBSC, administering only two immune suppressors can increase acute GVHD rates
**Carnevale-Schianca** [34]	MAC (Bu/Flu + others)	PT-Cy day +3, +4 and tacrolimus and MMF	MRD (#10) MUD (#25)	PBSC	Grade 2–4 17%	7%	77% at 2 years	This study proved that maintaining three immune suppressive drugs, GVHD incidence can remain low even if using MAC and PBSC
**Greco** [35]	MAC (Treo/Mel/Flu)	PT-Cy day +3, +4 and sirolimus and MMF (for MUD)	MRD (#15) MUD (#13)	PBSC	Grade 2–4 23%	13%	64% at 2 years (estimated)	This study showed that sirolimus can substitute tacrolimus with good results
**Bolaños-Meade** [36]	RIC (Flu/Cy/TBI)	PT-Cy day +3, +4 and tacrolimus and MMF	MRD (#29) MUD (#50) MMUD (#9)	PBSC	Grade 2–4 32%	39%	71% at 1 year	This study showed how PT-Cy/tacro/MMF is the most effective GVHD strategy outside of the CNI/MTX setting
Comparison studies of PT-Cy vs. standard GVHD prophylaxis in the settings of MRD, MUD or MMUD								
**Battipaglia G** [37] **(PT-Cy vs. ATG for MMUD donors)**	PT-Cy: MAC 50% ATG: MAC 50%	PT-Cy or ATG plus one or two immune suppressive drugs	PT-Cy = 93 patients ATG = 179 patients	PT-Cy: PBSC 91% ATG: PBSC 92%	Grade 3–4: 9% versus 19% (*p* = 0.04) in favor pf PT-Cy group	No differences	63% vs. 45% at 2 years (*p* < 0.5) in favor pf PT-Cy if patients in CR at trasnplant	PT-Cy has less grade 3–4 acute GVHD and higher survival than ATG-based GVHD prophylaxis
**Battipaglia G** [38] **(PT-Cy vs. ATG for MRD donors)**	PT-Cy: MAC 59% ATG: MAC 48%	PT-Cy or ATG plus one or two immune suppressive drugs	MRD PT-Cy = 197 patients ATG = 1913 patients	PT-Cy: PBSC 70% ATG: PBSC 95%	No differences	All grade in 37% and 30% (*p* = 0.02) in favor of ATG	No differences	PT-Cy is not superior to ATG when used in the MRD setting
Comparison studies of haploidentical, MUD and MMUD in the PT-Cy setting								
**Lorentino F** [39] **(MMUD 9/10 vs. MUD 10/10 in PT-Cy)**	MMUD: MAC 56% MUD: MAC 53%	MMUD: PT-Cy + CNI + MMF 68% MUD: PT-Cy + CNI + MMF 49%	MMUD = 159 MUD = 305	MMUD: PBSC 88% MUD: PBSC 88%	No differences	No differences	No differences	PT-Cy abrogates the effect of 1 HLA mismatch in the setting of 9 of 10 or 10 of 10 unrelated donors
**Gooptu M** [40] **(Haploidentical vs. MUD)**	Haploidentical: MAC 41% MUD: MAC 34%	PT-Cy + CNI + MMF	Haploidentical = 2036 MUD = 284	Haploidentical: PBSC 63% MUD: PBSC 85%	RIC cohort: lower acute GVHD in the MUD group	MAC cohort: lower chronic GVHD in the MUD group	RIC cohort: 54% versus 67% in favor of haploidentical cohort	MUD should be preferred over haploidentical donor when using PT-Cy

In the MAC-peripheral blood stem cell setting, alerts regarding increased acute GVHD severe toxicity have been reported. Mielcarek et al. used PT-Cy and cyclosporine (CSA) as well as two different MACs (busulphan/fludarabine and total body irradiation) for high-risk malignancies. Grade 2–4 acute GVHD was 77% at day +100 [33].

To obviate the increase in acute GVHD incidence, the other authors decided to maintain the original three-drug GVHD prophylaxis (PT-Cy + MMF + CSA/tacrolimus) in the HLA-matched donor setting as well. Carnevale-Schianca et al. used PT-CY + MMF + CSA in 35 patients with high-risk malignancies [34]. All-grade acute GVHD was 12%. At 2 years, non-relapse mortality, event-free survival, and overall survival were 3%, 54%, and 77%, respectively. Another study from Greco et al. that used PT-Cy + sirolimus with or without MMF showed that grade 2–4 acute GVHD was 23% at day +100 [35]. At the 1-year follow-up, TRM, relapse, and overall survival were 14%, 36%, and 64%, respectively. The use of sirolimus in combination with PT-Cy was also explored in a phase II study with a higher cumulative incidence of grade 2–4 acute and chronic GVHD of 46% and 31%, respectively, when compared to PT-Cy and CNI [41].

The prospective, randomized trial BMT-CTN 1301 recently presented final findings [42]. The study compared PT-Cy as the sole GVHD prophylaxis strategy with bone marrow graft to CD34+-selected peripheral blood stem cells and tacrolimus/methotrexate with bone marrow graft when using MAC and HLA-matched donors. The primary endpoint of this study was chronic GVHD/relapse-free survival. All three strategies resulted in acceptable outcomes. However, CD34+ selection was associated with worse survival when compared to the other two groups due to a higher TRM. PT-Cy alone had similar outcomes when compared to standard tacrolimus/methotrexate.

In the RIC setting, most patients received peripheral blood stem cell-derived grafts. In this regard, prospective trials have been undertaken or are ongoing. The BMT-CTN1203 study compared the use of PT-Cy/tacrolimus/MMF to tacrolimus/methotrexate/maraviroc or tacrolimus/methotrexate/bortezomib in a prospective, multicenter phase 2 trial [36]. The hazard-ratio for GVHD relapse-free survival was 0.72 (*p* = 0.044) for the PT-Cy/tacrolimus/MMF group. No differences were described in terms of relapse or survival. Thus, the PT-Cy-based GVHD prophylaxis was selected for the PROGRESS III trial, which aims to compare a PT-Cy strategy versus tacrolimus/methotrexate in the setting of HLA-matched donors [43]. Additionally, a prospective phase III HOVON-96 trial compared CSA+PT-Cy versus CSA+MMF in MRD and MUD peripheral blood stem cells [44]. Lower rates of acute and chronic GVHD were found in PT-Cy-based GVHD prophylaxis recipients. Preliminary results of a smaller phase 2 randomized study from Brissot et al. have also come to light [45]. In this trial, investigators compared PT-Cy/CSA/MMF to standard CSA/methotrexate + ATG GVHD prophylaxis for MUD alloHCT. Peripheral blood stem cell grafts and RIC were used in all patients. At the 6-month follow-up, grade 2–4 acute GHVD was 35% and 24% (*p* = 0.24) for PT-Cy and ATG cohorts, respectively. At the 1-year follow-up, all-grade chronic GVHD was 26% versus 30% (*p* = 0.56) for the same groups. No differences were observed in terms of non-relapse mortality, relapse, overall survival, GHVD relapse-free survival, or adverse events.

Finally, when comparing both MAC and RIC in the same analysis, results did not show a survival advantage with respect to conditioning intensity. A larger retrospective study from the EBMT of 423 patients with acute leukemia confirmed the acceptable results of PT-Cy in the HLA-matched setting using bone marrow or peripheral blood stem cell graft [46]. Bone marrow was the main graft source whenever sole PT-Cy was used (74% bone marrow versus 26% peripheral blood stem cells), while peripheral blood stem cells were the most frequent source if PT-Cy was used in combination with one or two immunosuppressive drugs. Both MAC and RIC regimens were used. Apparently, the only differences between the three groups (sole PT-Cy, PT-Cy + 1 drug, PT-Cy + 2 drugs) was an inferior all-grade chronic GVHD incidence whenever PT-Cy was associated with two immunosuppressive drugs. Another study from Nagler et al. compared PT-Cy alone to CSA/methotrexate in the setting of MRD donors for patients with acute myeloid leukemia [47]. Peripheral blood stem cell graft was the most common source used in this setting between the two groups. However, MAC conditioning was used less in the PT-Cy group. The only difference observed was a higher incidence of relapse in the PT-Cy group (HR 1.52; *p* = 0.02). A recent prospective, multicenter phase 2 trial for HLA-mismatched unrelated donor (UD) using PT-Cy/sirolimus/MMF and MAC (50%) or RIC (50%) conditioning showed the feasibility of alloHCT with mismatched UD [48]. All patients received bone marrow grafts. At the 1-year follow-up, overall survival was 76% with no significant differences between MAC/RIC and HLA match grade (7/8 versus 4–6 HLA matches). At day +100, acute GVHD incidence was 43% and 33% for the MAC and RIC groups, respectively. At the 1-year follow-up, chronic GVHD was 36% and 18% for the MAC and RIC groups, respectively. One-year non-relapse mortality and relapse were 8% and 30% for the MAC cohort, respectively, and 10% and 23% for the RIC cohort, respectively. Forty-eight percent of patients belonged to ethnic minorities, showing a possible way to expand access to alloHCT for patients with underrepresented ethnicities in international donor registries.

Only a few studies were reported in the mismatched UD setting. Gaballa et al. reported data regarding 46 patients undergoing RIC conditioning with bone marrow and receiving mismatched UD (9 of 10 HLA-matches) donors using PT-Cy/tacrolimus/MMF GVHD prophylaxis [49]. At day +100, grade 2–4 acute GVHD was 33%. At the 1-year follow-up, non-relapse mortality and overall survival were acceptable (34% and 47%). At the 2-year follow-up, all-grade chronic GVHD was 19%. Additionally, a recent study from Battipaglia et al. compared mismatched UD donor (9 of 10 HLA-matches) to haploidentical HCT using the same PT-Cy GVHD prophylaxis [50]. Interestingly, a lower frequency of HLA mismatches was associated with decreased leukemia-free survival; however, no significant differences in overall survival were observed. 

The use of simpler GVHD prophylaxis—comprising PT-Cy and tacrolimus for mismatched UD and MUD—has been reported in 109 patients by Pedraza et al. [51]. The authors described a similar cumulative incidence of grade 2–4 (31% vs. 32%) and 3–4 acute GVHD (9% vs. 7%) across the two groups. No differences in chronic GVHD, overall survival, TRM, and PFS were observed between the two groups, supporting that this approach could overcome the adverse effect by an HLA disparity.

When comparing PT-Cy to ATG within the context of mismatched UD and MRD, evidence from a retrospective registry study is available. In simple terms, PT-Cy appears to be superior to ATG when a 9 of 10 HLA-compatible donor is used in terms of survival [37]. However, this advantage is no longer present in the setting of MRD donors [38].

In conclusion, PT-Cy, in association with MMF and CNI, could represent a safe alternative to methotrexate/CNI in HLA-matched settings (Table 3). Nevertheless, higher-quality evidence is further required to confirm these preliminary data.

Additionally, the most recent studies compare results between different donors whenever a similar PT-Cy-based GVHD prophylaxis is used. In this context, retrospective registry study evidence shows that MUD donors should be preferred to haploidentical donors as well [40]. Among 9 of 10 and 10 of 10 unrelated donors, PT-Cy appears capable of abrogating differences between these two types of donors with respect to survival outcomes [39].

## 6. Future Perspectives

In the future, final results from prospective clinical trials will establish which GVHD prophylaxis is the most optimal, especially in the HLA-matched donor setting. At the moment, there is an absence of high-quality evidence to support the claim that this approach would be superior to standard GVHD prophylaxis (Table 3). PT-Cy has the potential to overcome HLA barriers and facilitate the possibility of faster immune tolerance [55]. Considering that disease relapse remains the first cause of death after alloHCT, it is important to develop more novel post-transplant strategies capable of minimizing the risk of relapse. Immunotherapies relying on an adaptive immune system may be used in the future to consolidate or maintain disease response after alloHCT. GVHD prophylaxis such as that of PT-Cy can induce faster immune tolerance and thereby allow for rapid suspension of immune suppressors. In other words, clinicians may be able to use immune therapies in a timely manner that could effectively reduce disease relapse after alloHCT.

## 7. Conclusions

PT-Cy-based GVHD prophylaxis has rapidly changed the therapeutic scenario in the alloHCT setting. While its efficacy in the haploidentical setting has been reproduced worldwide, its use in the HLA-matched setting should be considered experimental. Promising results have been prospectively obtained when PT-Cy is used in the MUD or mismatched UD setting. PT-Cy appears not to be inferior to standard CNI/methotrexate GVHD prophylaxis in this setting. However, when using MAC conditioning and peripheral blood stem cells as the graft source, the incidence of chronic GVHD is not so different from that observed with standard prophylaxis. In the MRD setting, the advantages of PT-Cy versus standard therapy are even less clear. The final results of larger, prospective multicenter clinical trials such as the PROGRESS III trial could resolve this open matter and determine whether PT-Cy-based GVHD prophylaxis is better than that which is methotrexate-based.

## Figures and Tables

**Table 1 jcm-10-03548-t001:** Selection of studies using PT-Cy in the setting of haploidentical donor alloHCT. GVHD = graft-versus-host disease; NRM = non-relapse mortality; OS = overall survival; HL = Hodgkin lymphoma; Flu = fludarabine; Cy = cyclophosphamide; TBI = total body irradiation; Thio = thiotepa; RIC = reduced-intensity conditioning; AML = acute myeloid leukemia; MDS = myelodysplasia; Mel = melphalan; Bu = busulfan; MAC = myeloablative; ALL = acute lymphoblastic leukemia; AA = aplastic anemia; MUD = matched-unrelated donor; Haplo = haploidentical; NHL = non-Hodgkin lymphoma; ATG = anti-thymocyte globulin.

Study	Disease	Number of Patients	Type of Conditioning	Grade 2–4 Acute GVHD	All-Grade Chronic GVHD	Relapse/NRM	OS
**Castagna** [13]	HL	62	Flu/Cy/TBI, Thio/Flu/Cy/TBI	23%	16%	21%/20% at 1 year	63% at 1 year
**Gauthier** [14]	HL	34	Flu/Cy/TBI	28%	15%	28%/12% at 3 years	78% at 3 years
**Martinez** [15]	HL	98	RIC (90%)	33%	26%	39% at 2 years/17% at 1 year	67% at 3 years
**Ciurea** [16]	AML/MDS	43	Flu/Mel based	35%	9%	24%/34% at 2 years	42% at 2 years
**Gayoso** [17]	AML/MDS	64	Bu/Flu/Cy (MAC)	29%	28%	25%/19% at 2 years	56% at 2 years
**Shem-Tov** [18]	ALL	136	RIC, MAC	28%	44%	28%/23% at 3 years	54% at 3 years
**Santoro** [19]	ALL	208	RIC, MAC	31%	29%	37%/32% at 3 years	33% at 3 years
**Prata** [20]	AA	33	RIC	23%	20%	------	78% at 2 years
Comparison studies of haploidentical donor alloHCT versus 8 of 8 HLA-matched unrelated donor with standard GVHD prophylaxis							
**Ciurea** [5]	AML	Haplo = 192 MUD = 1982	Haplo = MAC 104; RIC 88 MUD = MAC 1245; RIC 737	MAC setting: 16% vs. 33% (*p* < 0.0001) RIC setting: 19% versus 28% (*p* = 0.05)	MAC setting: 30% vs. 53% (*p* < 0.0001) RIC setting: 34% versus 52% (*p* = 0.002)	No differences between haplo and MUD	MAC setting: 45% vs. 50% (*p* = 0.38) RIC setting: 46% versus 44% (*p* = 0.71)
**Kanate** [4]	NHL and HD	Haplo = 185 MUD without ATG = 241 MUD with ATG = 491	RIC for all patients	Grade III-IV acute GVHD 8%, 12%, 17%	13%, 51% and 44% (*p* < 0.001)	No differences between haplo and MUD	60%,62%,50% (*p* = 0.02)

**Table 3 jcm-10-03548-t003:** Post-transplant cyclophosphamide indications and levels of evidence in relation to different donor settings. Levels of evidence are based on Oxford Center for Evidence-based Medicine (OCEBM) Levels of Evidence Working group (2009) [52].

	Haploidentical	MUD	MMUD	MRD
Prospective studies	PT-Cy is strongly suggested in this setting due to its efficacy and easy-to-use characteristics compared to other haploidentical strategies	PROGRESS III study is currently comparing in a phase 3 trial if PT-Cy/tacro/MMF is superior to standard tacro/MTX (without ATG) in the RIC/PBSC setting [43]. Progress II study showed that PT-Cy GVHD prophylaxis alone is not superior to tacro/MTX (without ATG) in the MAC/BM setting [42]. A recent phase 2 randomized trial did not show any differences between PT-Cy/CSA/MMF and CSA/MTX + ATG in this setting [45].	Progress III study is currently comparing in a phase 3 trial if PT-Cy/tacro/MMF is superior to standard tacro/MTX in the RIC/PBSC setting [43]. One prospective phase 2 trial showed feasibility of PTCy/sirolimus/MMF in the setting of MMUD when using BM graft [48].	Progress III study is currently comparing in a phase 3 trial if PT-Cy/tacro/MMF is superior to standard tacro/MTX in the RIC/PBSC setting [43]. Progress II study showed that PT-Cy GVHD prophylaxis alone is not superior to tacro/MTX (without ATG) when using MAC/BM setting [42]. A recent phase 2 randomized trial did not show any differences between PT-Cy/CSA/MMF and CSA/MTX + ATG in this setting [45].
Retrospective studies	Retrospective studies confirm the effectiveness of this strategy in large cohorts of patients (Table 1)		A MMUD donor should be preferred over a haploidentical one when using PT-Cy-based strategy [53].	PT-Cy prophylaxis alone was not shown to be superior to CSA/MTX in this setting [47].
Final recommendations	In support of PT-Cy-based strategies due to reduced acute and chronic GVHD and favorable toxicity profile when compared to other haploidentical strategies.Level of evidence 2a	PT-Cy-based strategies could be a clinical option, especially in the RIC/BM setting [4,5]. When using MAC/PBSC schemes and compared to ATG-based strategies, advantages are less clear in terms of GVHD incidence. Apparently, using a MUD donor is better than a haploidentical one when using PT-Cy-based strategies in the setting of myeloid diseases and RIC conditioning [40]. Level of evidence 3a (waiting for randomized clinical trial results).	PT-Cy-based strategies could be a clinical option, especially in the context of RIC/BM setting. When using MAC/PBSC schemes and compared to ATG-based strategies, advantages are less clear in terms of GVHD incidence. Level of evidence 3a (waiting for randomized clinical trial results).	Current retrospective data do not show a significant advantage over CNI-based strategies in this setting [54]. Level of evidence 3b (waiting for randomized clinical trial results).

## Data Availability

Not applicable.

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
