# Peer review of "Is Post-Transplant Cyclophosphamide the New Methotrexate?"

_jcm, 2021, doi:10.3390/jcm10163548_

Round 1
Reviewer 1 Report
In their manuscript entitled: “Is Post-transplant Cyclophosphamide the New Methotrexate?”, the authors aim to overview current literature on the use of PT-Cy in the setting of HLA-matched HCT, after its relative success in haplo-HCT. This is a relevant and interesting topic to the current developments in the field, but the manuscript fails to provide a comprehensive overview of the literature, and does not meet up to its aim. There are some major issues that need to be addressed.
While the tables in this review provide some overview, it is unclear whether these are complete, or just provide an overview of a selection of studies. Authors made tables from “a selection of studies” and “a list of relevant studies”: why are these studies selected or considered relevant? This should be described in text, or all studies in this setting should be included.
For the manuscript to provide a comprehensive overview of the comparison between PT-Cy and other GvHD-prophylactic approaches, the tables should include outcomes from both and not only PT-Cy.
In text, the authors describe each separate “selected” study, but comparisons are not clear (or even made). It would be helpful if the outcome data are described in a somewhat more summarized fashion per setting, and then compared to other prophylactic approaches.
Line 43: the authors state that the efficacy of post-Cy is through immune tolerance and rapid immune reconstitution, but refer to a review from the authors themselves. Please provide the primary references. Furthermore, this is a highly relevant statement since other studies report on delayed immune reconstitution after haplo-HCT, which therefore should be elaborated more in text.
In the section of therapeutic strategies after transplant, only DLI is discussed. It is not clear what the aim would be of these therapeutic strategies and what the point of this paragraph is. Authors should clarify and include discussion of other strategies as well.
Author Response
Reviewer #1
Comments and Suggestions for Authors
In their manuscript entitled: “Is Post-transplant Cyclophosphamide the New Methotrexate?”, the authors aim to overview current literature on the use of PT-Cy in the setting of HLA-matched HCT, after its relative success in haplo-HCT. This is a relevant and interesting topic to the current developments in the field, but the manuscript fails to provide a comprehensive overview of the literature, and does not meet up to its aim. There are some major issues that need to be addressed.
 
While the tables in this review provide some overview, it is unclear whether these are complete, or just provide an overview of a selection of studies. Authors made tables from “a selection of studies” and “a list of relevant studies”: why are these studies selected or considered relevant? This should be described in text, or all studies in this setting should be included.
We agree with the reviewer that our review is not reporting all the studies in which PT-Cy was used. This topic would deserve a complete textbook. However, the aim of this review was to summarize current topics such as the use of PT-Cy in the HLA-identical context.
We specify in the next why these studies were considered relevant.
“Study selection was made prioritizing the type of study (prospective over retrospective), population size and date of publication.”(Methods section)
 
For the manuscript to provide a comprehensive overview of the comparison between PT-Cy and other GvHD-prophylactic approaches, the tables should include outcomes from both and not only PT-Cy.
In text, the authors describe each separate “selected” study, but comparisons are not clear (or even made). It would be helpful if the outcome data are described in a somewhat more summarized fashion per setting, and then compared to other prophylactic approaches.
We agree with the reviewer that comparison studies represent an important topic to be addressed in the manuscript. We updated the manuscript and also the table with the most relevant comparison study of PT-Cy versus ATC and also different types of donors in the PT-Cy context.
 
Line 43: the authors state that the efficacy of post-Cy is through immune tolerance and rapid immune reconstitution, but refer to a review from the authors themselves. Please provide the primary references. Furthermore, this is a highly relevant statement since other studies report on delayed immune reconstitution after haplo-HCT, which therefore should be elaborated more in text.
We thank the reviewer for this comment. We insert the original studies references. However, we do not consider it appropriate to expand this immunological topic in a clinical review due to the readership target and the limited space available in the review.
 
In the section of therapeutic strategies after transplant, only DLI is discussed. It is not clear what the aim would be of these therapeutic strategies and what the point of this paragraph is. Authors should clarify and include discussion of other strategies as well.
We agree with the reviewer that this section could be out of target for the review. We decided to remove it from the paper to let more space for transplant-related issues.
Reviewer 2 Report
Mussetti et al have written a comprehensive summary on the current knowledge status for use of PT-Cy in alloHCT. The intention and methods of the review is clear and generally the manuscript has a good structure. I have some comments on the presentation.
- Since the title highlights the comparison between Cyclophosphamide and Methotrexate I think Methotrexate as a drug and its use in alloHCT should be presented better in the Introduction to explain why this is an important and relevant question. It would also be clarifying to briefly present all the GVHD prophylaxis drugs that are later mentioned and compared in the later sections.
- The language should be checked and edited by someone with English as their mother tongue. Some sentences suffer from phrasing that makes them a bit unclear. An example from the intro on Cyclophosphamide: "Biologically, its efficacy is related to a strong induction of immune tolerance through peripheral and central mechanisms and also a rapid immune reconstitution". Exchanging "also" for "to", makes the sentence acceptable.
- Abbreviations are used very frequently. It is often a routine to abbreviate but it makes reading less smooth. The authors should go over the used abbreviations and only keep those that are used so many times it makes sense (more than 3?)
Author Response
Reviewer #2:
- Since the title highlights the comparison between Cyclophosphamide and Methotrexate I think Methotrexate as a drug and its use in alloHCT should be presented better in the Introduction to explain why this is an important and relevant question. It would also be clarifying to briefly present all the GVHD prophylaxis drugs that are later mentioned and compared in the later sections.
We expand the introduction section with a brief historical part regarding the use of methotrexate and ATG as GVHD prophylaxis strategies. We believe this could help the reader in understanding a little more the evolution of the GVHD prophylaxis evolution.
- The language should be checked and edited by someone with English as their mother tongue. Some sentences suffer from phrasing that makes them a bit unclear. An example from the intro on Cyclophosphamide: "Biologically, its efficacy is related to a strong inductionof immune tolerance through peripheral andcentral mechanisms and also a rapid immune reconstitution". Exchanging "also" for "to", makes the sentence acceptable.
An English Professor reviewed the manuscript as suggested
- Abbreviations are used very frequently. It is often a routine to abbreviate but it makes reading less smooth. The authors should go over the used abbreviations and only keep those that are used so many times it makes sense (more than 3?)
Thank you for this comment. We modified the manuscript to avoid the use of unnecessary abbreviations.
Round 2
Reviewer 1 Report
No further comments. Previous comments adequately tackled.